# Prediction of Sludge Volume Index in a Wastewater Treatment Plant Using Recurrent Neural Network

Praewa Wongburi [1,*] and Jae K. Park [2]

1   Faculty of Environment and Resource Studies, Mahidol University, Nakhon Pathom 73170, Thailand
2   Department of Civil and Environmental Engineering, University of Wisconsin-Madison,
    Madison, WI 53706, USA; jkpark@wisc.edu
*   Correspondence: praewa.won@mahidol.ac.th

**Abstract:** Sludge Volume Index (SVI) is one of the most important operational parameters in an activated sludge process. It is difficult to predict SVI because of the nonlinearity of data and variability operation conditions. With complex time-series data from Wastewater Treatment Plants (WWTPs), the Recurrent Neural Network (RNN) with an Explainable Artificial Intelligence was applied to predict SVI and interpret the prediction result. RNN architecture has been proven to efficiently handle time-series and non-uniformity data. Moreover, due to the complexity of the model, the newly Explainable Artificial Intelligence concept was used to interpret the result. Data were collected from the Nine Springs Wastewater Treatment Plant, Madison, Wisconsin, and the data were analyzed and cleaned using Python program and data analytics approaches. An RNN model predicted SVI accurately after training with historical big data collected at the Nine Spring WWTP. The Explainable Artificial Intelligence (AI) analysis was able to determine which input parameters affected higher SVI most. The prediction of SVI will benefit WWTPs to establish corrective measures to maintaining stable SVI. The SVI prediction model and Explainable Artificial Intelligence method will help the wastewater treatment sector to improve operational performance, system management, and process reliability.

**Keywords:** Sludge Volume Index; recurrent neural networks; Explainable Artificial Intelligence; Wastewater Treatment Plant; time-series data; prediction model

## 1. Introduction

The activated sludge process has been used worldwide for the biological wastewater treatment system [1]. One of the critical steps is to separate activated sludge from liquid before discharge. Thus, SVI (mL/g), an indicator of solid separation, is an important operational parameter. It is defined as 'the volume (in mL) occupied by 1 g of activated sludge after settling the aerated liquid for 30 min [2]. Many researchers have developed the SVI model using Artificial Neural Networks (ANNs) [3–5]. Unfortunately, there are no deterministic explainable models to predict SVI and interpret the result due to the unexplainable behavior of microorganisms causing the sludge bulking (settling) problem and uninterpretability of the neural network model. A common problem in the activated sludge system is poor solid separation at the secondary clarification stage [6]. Excess growth of filamentous organisms makes activated solids difficult to settle in secondary clarifiers, leading to a potential violation of the total solids regulatory limit. The recurrent neural network (RNN) model was used to predict SVI from big data generated at the Nine Springs Wastewater Treatment Plant (WWTP), Madison, Wisconsin. The model will aid in predicting potential settling issues and providing possible reasons for higher SVI prediction. This model will significantly enhance the activated sludge system performance in WWTPs.

Control of the activated sludge process is difficult for many WWTPs due to the complexity of the biological and chemical reactions and variations in the influent water quality and flow rate [7]. The activated sludge process control can be improved by evaluating the

causes of higher SVI and taking preventive measures. Several variables can impact the settleability of sludge in the clarifiers, such as filamentous bacteria, rain events, and water temperature. Filamentous bacteria such as *Sphaerotilus natans* (*S. natans*) and *Microthrix parvicella* (*M. parvicella*) impact sludge settleability because these bacteria create more buoyant flocs [8]. Moreover, a balance between floc- and filamentous-forming bacteria is required. There are still many causes leading to problems in the activated sludge process. Therefore, sludge settling problems occurring in the activated sludge process can be avoided by taking the right action through the real-time monitoring system, Explainable Artificial Intelligence (AI) algorithms, and proactive measures.

In recent years, predictive modeling approaches have been increasingly applied in many industries [9]. Modeling with RNNs is a current trend in deep learning neural networks algorithms [10]. The advantage of RNN algorithms is the capability to handle sequential data with a variable dataset [11]. However, the more accuracy of the model like RNN, makes the model utmost difficult to interpret [12]. SVI is one of the most important parameters that monitors the activated sludge settling performance in WWTPs. RNN models can be used to predict SVI in the activated sludge system with a validated dataset and then applied to an explainable function to interpret the result, allowing operators to take preventive measures before the sludge settling issue arises during the operation of the activated sludge process.

### 1.1. Sludge Volume Index (SVI)

Operators use SVI to determine and compare mixed liquor settleability [13]. It mathematically relates settled sludge volume in the settleometer to mixed liquor suspended solids (MLSS) concentration. SVI relates sludge volume in milliliters to MLSS concentration in grams per liter as follows [14]:

$$SVI \ (\text{mL/g}) = \frac{Settled \ Sludge \ Volume \ SSV_{30} \ (\text{mL/L})}{MLSS \ (\text{mg/L})} \times 1000 \ \text{mg/g} \tag{1}$$

where SSV30 (in units of milliliters per liter) is the volume of sludge that settles in a graduated cylinder of mixed liquor in 30 min and mixed liquor suspended solids (MLSS) (mg/L) is the MLSS concentration in aeration basins. The common range for an SVI at a conventional activated sludge system is between 50 and 150 [15]. Optimum SVI must be determined for each WWTP experimentally. SVI is an excellent indicator of the settling characteristics of the sludge. However, SVI varies with the characteristics and concentration of the mixed-liquor solids. Thus, observed values at a given WWTP should not be compared with those reported for other plants or in the literature. Typical SVI values for a good settling sludge with 1500–3500 mg/L of mixed liquor concentrations range from 80 to 120 [15]. Filamentous bulking increases SVI even if the MLSS concentration is the same. Therefore, SVI is a good indicator of filamentous bulking.

### 1.2. Activated Sludge Process

The activated sludge process is a biological wastewater treatment process where microorganisms biodegrade organics present in wastewater as a carbon source. The settleability of the activated sludge depends on the size, density, and shape of the flocs and the competency of the secondary clarifier. Settleability can be affected by the extent of the filamentous bacteria population. These bacteria can form strings as they grow rather than forming flocs. Excess growth of these filamentous organisms can cause a bulking condition, resulting in poor settling and taking up more sludge blanket volume in the secondary clarifier. This condition may be triggered by several factors, such as inadequate dissolved oxygen (DO) and nutrient imbalance, leading to solids loss in the clarifier effluent due to poor solid separation [16]. Therefore, the control of sludge bulking is crucial in the activated sludge process.

### 1.3. Filamentous Bulking

Filamentous bulking is the number one cause of effluent non-compliance in the United States (U.S.). Filamentous bulking and foaming are serious issues in an activated sludge operation, affecting most WWTPs [16]. A bulking sludge settles slowly and does not settle compactly, causing subsequent solids overflow at the secondary clarifier. An operational target SVI often used for operation is <150 mL/g, although each WWTP has unique SVI values for safe operation, varying from <100 mL/g to >300 mL/g, depending on the hydraulic considerations and the capacity and performance of the secondary clarifier. For example, a bulking sludge may be acceptable if the secondary clarifier is sufficiently large.

### 1.4. Recurrent Neural Network (RNN)

The RNN is a natural generalization of feedforward neural networks to sequences. Given a sequence of inputs $(x_1, \dots, x_t)$, a standard RNN computes a series of outputs $(y_1, \dots, y_t)$ by iterating the following equation [17]:

$$h_t = sigm \left( W^{hx} x_t + W^{hh} h_{t-1} \right) y_t = W^{yh} h_t \qquad (2)$$

The RNN can easily map sequences to sequences whenever the alignment between the inputs and the outputs are known ahead of time. However, it is unclear how to apply an RNN to problems whose input and output sequences have different lengths with complicated and non-monotonic relationships. In Figure 1 [18], an RNN, A, has an input $x_t$ and output $h_t$. A loop allows information to persist and pass from one step of the network to the next, in which traditional neural networks cannot handle this.

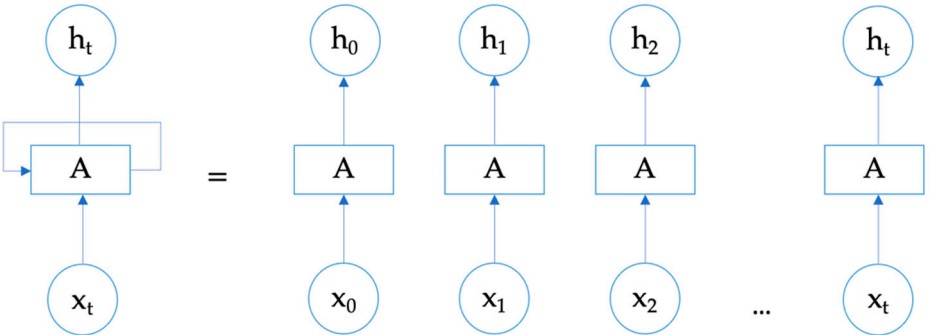

**Figure 1.** A Recurrent Neural Networks. (Source: Olah, 2015).

The most straightforward strategy for general sequence learning is to map the input sequence to a fixed-sized vector using one RNN and then map the vector to the target sequence with another RNN.

### 1.5. Shortcomings of Previous Predictive Models

Monitoring and controlling wastewater treatment processes have increased consideration and led to several models for the biological treatment processes in WWTPs such as Activated Sludge Models (ASM1, ASM2, ASM2d, and ASM3) [19]. Still, the complex structures of these models involving large numbers of parameters that must be identified make them inappropriate for monitoring purposes [20]. For instance, the model ASM1 contained 13 non-linear differential equations and 19 parameters, which are very difficult for computation [21].

Harrou et al. (2018) have studied a monitoring strategy using deep learning approaches, deep belief networks (DBNs), and a one-class support vector machine (OCSVM). However, when the data are highly noisy, false alarms might be generated during the fault detection task. As a result, deeper learning algorithms such as deep neural networks (DNNs), artificial neural networks (ANNs), and recurrent neural networks (RNNs) should be proposed to achieve complex input information in WWTPs.

Many researchers have recently developed various types of predictive models. Zhao et al. (2020) found that with the development of AI technology, the number of research publications of AI application to wastewater treatment was 19 times greater in 2019 than in 1995, and papers had 36 more citations on average. However, most AI applications in WWTP are limited to ANN models. RNN models have not so far been applied to WWTP operations.

### 1.6. Explainable Artificial Intelligence

As predictive algorithms play an important role in our lives, they become increasingly complex. Explaining why an algorithm makes certain decisions is ever more crucial [22]. Accuracy and interpretability are two main factors of successful predictive models. Typically, a decision must be made in favor of complex black box models such as Recurrent Neural Networks (RNN) for accuracy versus less accurate but more interpretable traditional models such as the logistics regression model [23]. However, the highest accuracy for big datasets is often attained by complex models that experts struggle to explain. Several approaches have been developed to help users understand the predictions of complex models. Still, it is often unclear how these approaches are related and when one technique is preferable over another [24].

SHAP (SHapley Additive exPlanation) values are a unified measure of feature importance [24]. In other words, SHAP assigns each input/feature an importance value for a particular prediction. They interpret how to get from the base value $E[f(z)]$ that would be forecasted when the unknown featured to the current output $f(x)$ occurred. Figure 2 shows the diagram for a single ordering. The SHAP values start from averaging the $-i$ values through all possible orderings. The calculation of SHAP values determine the importance of a feature by comparing what a model predicts with and without feature. The calculation is made in every possible order because the order of a model can affect a prediction. Therefore, it is suitable for non-linear or complex time-series data, which the order of data affects a prediction output.

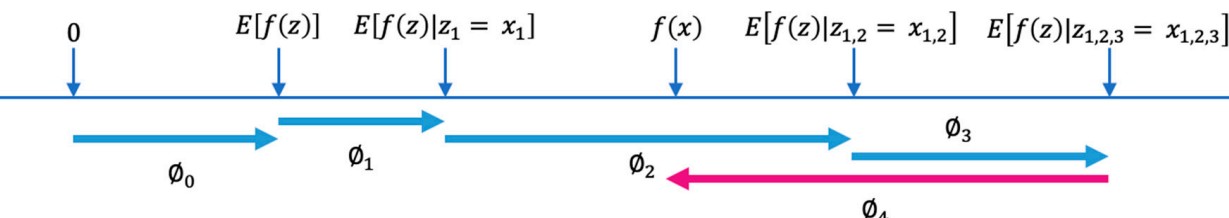

**Figure 2.** The diagram that shows a single ordering. (Source: Lundberg & Lee, 2017.).

## 2. Materials and Methods

### 2.1. Data Collection

The dataset from 1997 to 2020 was obtained from the Nine Springs Wastewater Treatment Plant (Nine Springs WWTP), Madison, Wisconsin, in the U.S. Figure 3 summarizes the data from the Nine Springs WWTP. There are 30 columns of parameters, including flow rate, influent parameters, effluent parameters, SVI, sludge age, and 8642 rows of data after removing missing rows.

The next step is to select the inputs of the dataset. In this study, flow rate, influent Biochemical Oxygen Demand ($BOD_5$), Total Suspended Solids (TSS), Total Kjeldahl Nitrogen (TKN), Ammoniacal nitrogen ($NH_3N$), Total Phosphorus (TP), and organic loading (flow rate $\times$ influent $BOD_5$) were selected as inputs of the model, and the output is SVI. Then, data visualization was applied using the Python program. Figure 4 displays Sludge Volume Index (SVI) data from 1996 to 2020.

```
<class 'pandas.core.frame.DataFrame'>
Index: 3738 entries, 2010-01-01 to 2020-03-05
Data columns (total 30 columns):
 #   Column               Non-Null Count   Dtype
---  ------               --------------   -----
 0   Flow_In (MGD)        3738 non-null    float64
 1   Influent_BOD5        3738 non-null    float64
 2   Influent_CBOD5       3738 non-null    float64
 3   Influent_TSS         3738 non-null    float64
 4   Influent_TKN         3738 non-null    float64
 5   Influent_NH3-N       3738 non-null    float64
 6   Influent_TP          3738 non-null    float64
 7   Influent_VSS         3738 non-null    float64
 8   Influent_pH          3738 non-null    float64
 9   Flow_Eff (MGD)       3738 non-null    float64
 10  Effluent_BOD5        3738 non-null    float64
 11  Effluent_TSS         3738 non-null    float64
 12  Effluent_TKN         3738 non-null    float64
 13  Effluent_NH3-N       3738 non-null    float64
 14  Effluent_TP          3738 non-null    float64
 15  Effluent_VSS         3738 non-null    float64
 16  Effluent_Temp        3738 non-null    float64
 17  Effluent_DO          3738 non-null    float64
 18  SVI_Plant1           3738 non-null    float64
 19  SVI_Plant2           3738 non-null    float64
 20  SVI_Plant3           3738 non-null    float64
 21  SVI_Plant4           3738 non-null    float64
 22  SludgeAge_Plant1     3738 non-null    float64
 23  SludgeAge_Plant2     3738 non-null    float64
 24  SludgeAge_Plant3     3738 non-null    float64
 25  SludgeAge_Plant4     3738 non-null    float64
 26  SludgeAge_Plant1_AO  3738 non-null    float64
 27  SludgeAge_Plant2_AO  3738 non-null    float64
 28  SludgeAge_Plant3_AO  3738 non-null    float64
 29  SludgeAge_Plant4_AO  3738 non-null    float64
dtypes: float64(30)
memory usage: 905.3+ KB
```

**Figure 3.** The data from the Nine Spring Wastewater Treatment Plant.

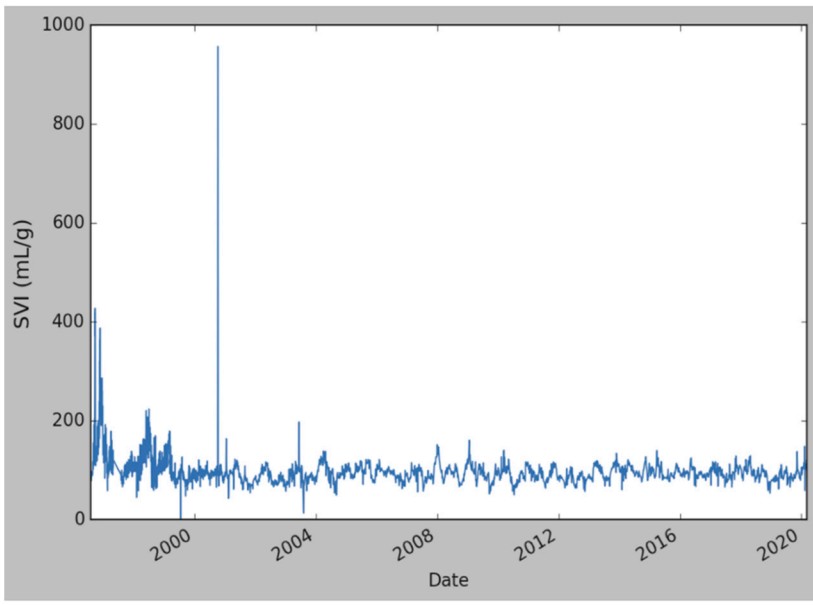

**Figure 4.** Sludge Volume Index (SVI) data from 1996 to 2020.

The next step, box plot analysis, was used to analyze the yearly data in Figure 5. The plot shows that in 2000, there are many errors in SVI data. Therefore, the dataset from 2001 to 2020 was used. Figure 6 shows SVI data from 2001 to 2020.

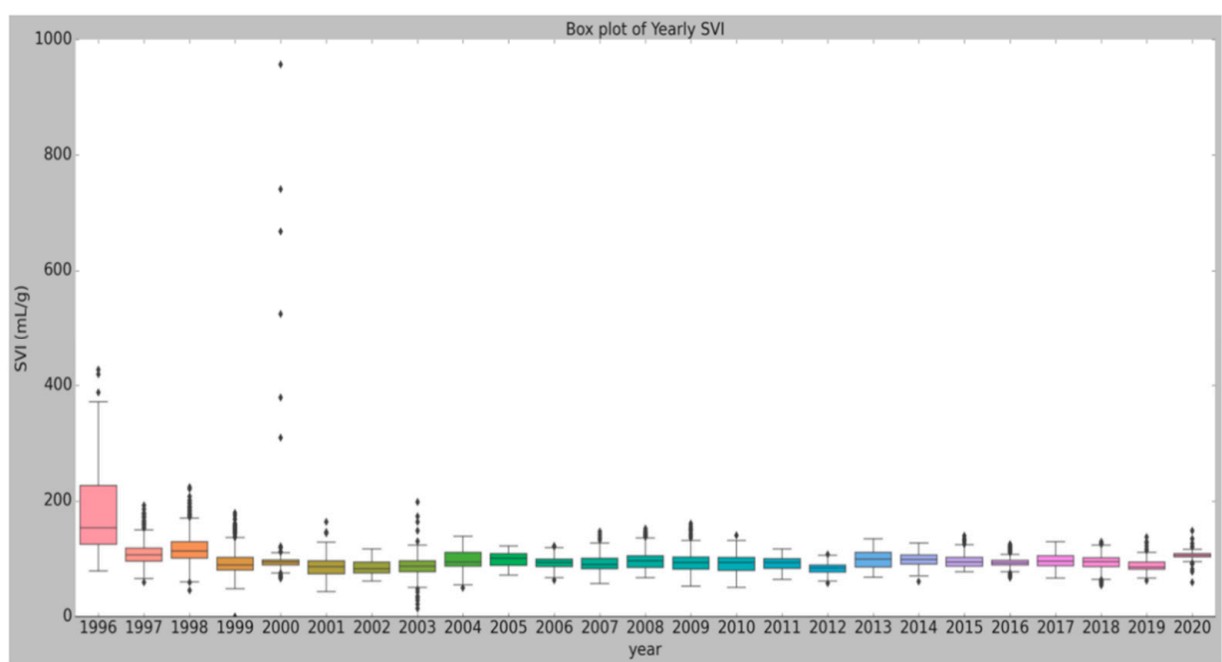

**Figure 5.** Box plot of yearly SVI from 1996 to 2020.

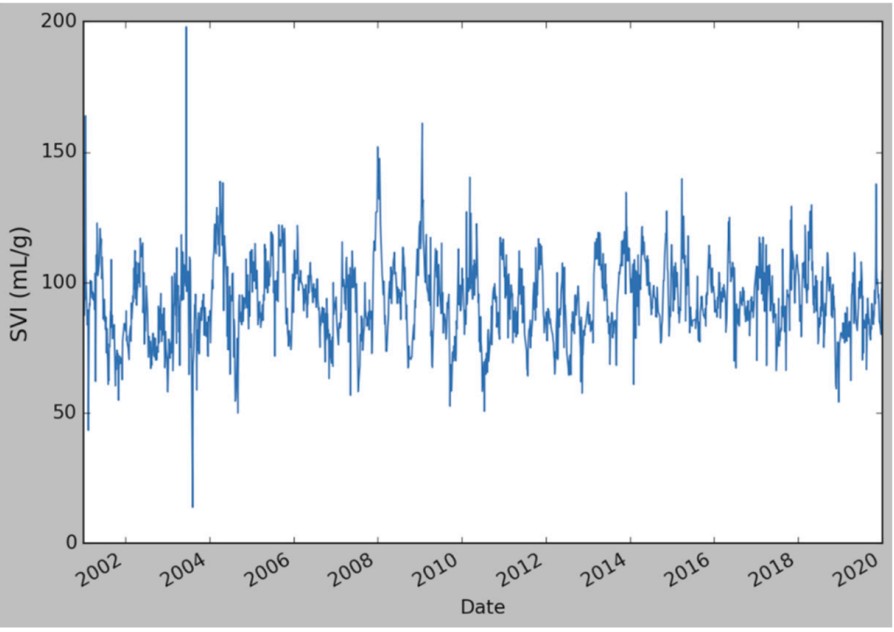

**Figure 6.** Sludge Volume Index (SVI) data from 2001 to 2020.

Figure 6 shows that the maximum value of SVI data decreases from 1000 to 200 mL/g. SVI data from 2001 to 2020 are plotted in Figure 7. Therefore, the dataset was more stable and had fewer errors. In this box plot, 2001, 2003, 2008, and 2009 have a wide range of SVI values, implying that the process was unstable due to the plant modification at that time.

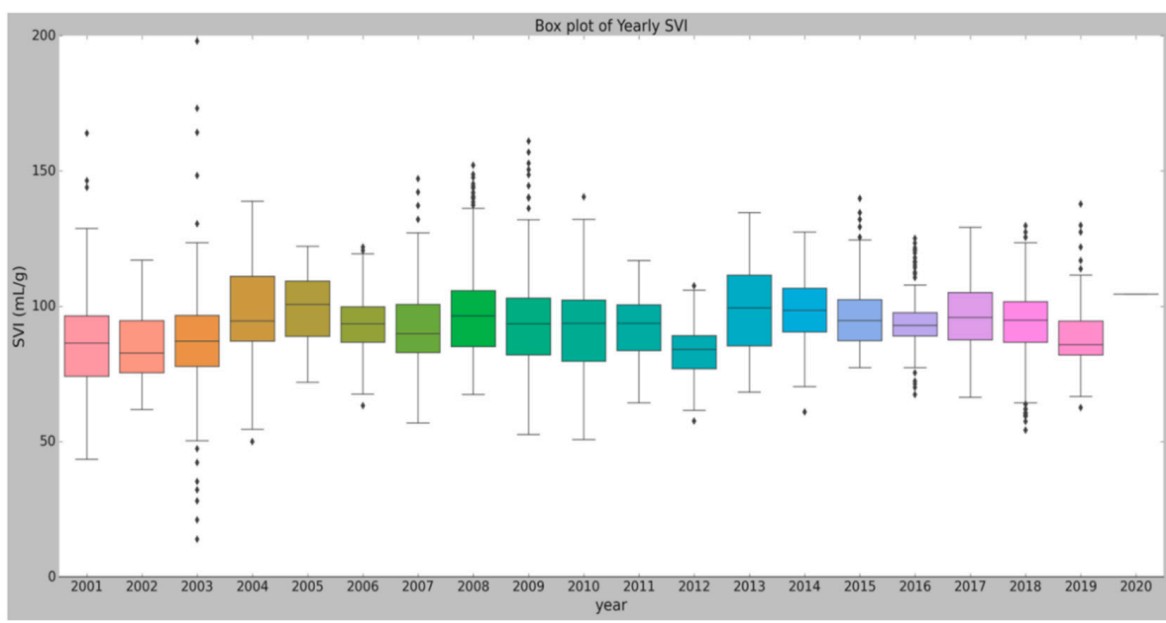

**Figure 7.** Box plot of yearly SVI from 2001 to 2020.

Therefore, the third dataset was selected from 2010 to 2020, which shows more stable data with the appropriate range of SVI of 50 to 150 mL/g. Figure 8 shows the SVI time-series data from 2010 to 2020.

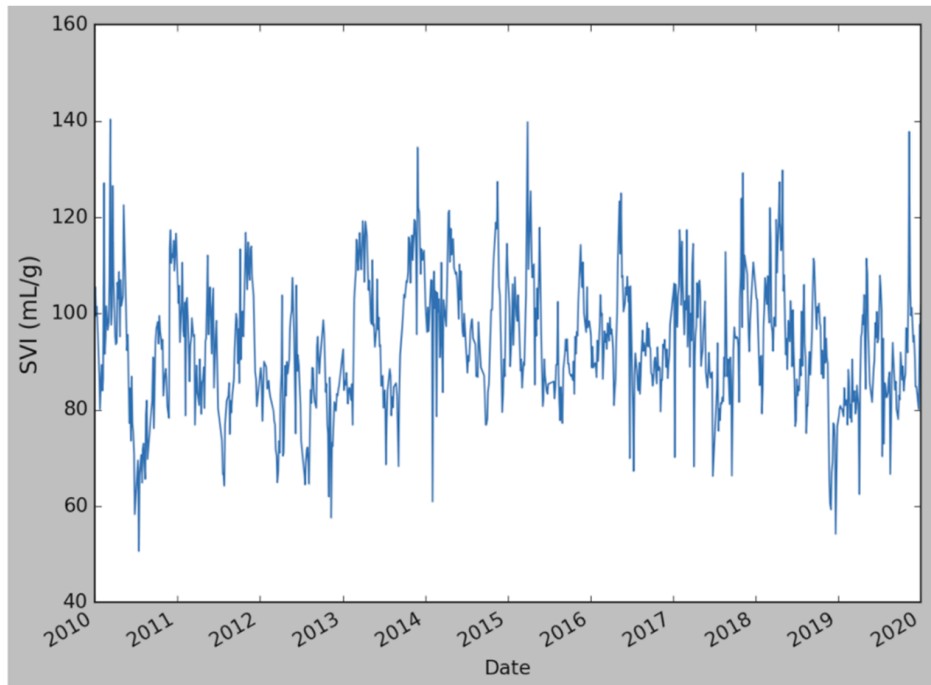

**Figure 8.** Sludge Volume Index (SVI) data from 2010 to 2020.

*2.2. Recurrent Neural Networks Models and Shapley Explanation*

The recurrent neural networks models were selected, and the inputs and output values were normalized (from 0–1) in Figure 9. Then, the dataset was separated into training (80% of the dataset) and testing (20% of the dataset) sets and put in the model. The last step is to apply the Shapley function to interpret the prediction result.

|   | var1(t-1) | var2(t-1) | var3(t-1) | var4(t-1) | var5(t-1) | var6(t-1) | \ |
|---|-----------|-----------|-----------|-----------|-----------|-----------|---|
| 1 | 0.631085 | 0.192411 | 0.003406 | 0.563376 | 0.563376 | 0.369318 | |
| 2 | 0.621351 | 0.078167 | 0.014519 | 0.423265 | 0.423265 | 0.306818 | |
| 3 | 0.611617 | 0.005066 | 0.004602 | 0.364614 | 0.364614 | 0.332386 | |
| 4 | 0.601882 | 0.076612 | 0.013616 | 0.423265 | 0.423265 | 0.318182 | |
| 5 | 0.589009 | 0.066292 | 0.028674 | 0.387423 | 0.387423 | 0.315341 | |

|   | var7(t-1) | var8(t-1) | var9(t-1) | var1(t) |
|---|-----------|-----------|-----------|---------|
| 1 | 0.734654 | 0.783042 | 0.591954 | 0.621351 |
| 2 | 0.645545 | 0.783042 | 0.568966 | 0.611617 |
| 3 | 0.681188 | 0.783042 | 0.571839 | 0.601882 |
| 4 | 0.546535 | 0.750624 | 0.540230 | 0.589009 |
| 5 | 0.534653 | 0.678304 | 0.581897 | 0.576135 |

**Figure 9.** Normalization inputs and output values of the models.

### 3. Results

The first dataset is from 1996 to 2020, the original dataset from the Nine Springs WWTP. After data analysis, the second and third datasets were created. The second dataset is the dataset between 2001 and 2020 because the data in 2000 was found to have a significant error in the dataset. The third dataset is the data from 2010 to 2020, in which the out-of-range (50 to 150 mL/g) SVI data were removed. The result shows that the second and third datasets are more suitable for applying in the model. Figure 10 shows the normal distribution used to determine if the data distribution departs from the normal distribution. Kurtosis and Skewness tests were also calculated. Figure 10a shows that the Kurtosis of the first dataset was far from 0, implying that the distribution had heavier tails. Skewness measures the asymmetry of the distribution. If the skewness is between −0.5 and 0.5, the data are symmetrical. If the skewness is between −1 and 0.5 or between 0.5 and 1, the data are moderately skewed. If the skewness is less than −1 or greater than 1, the data are highly skewed. Therefore, the first dataset was very highly skewed.

Data visualization was performed to determine the appropriateness of the dataset. The data in 2000 appears to have a high error. Thus, the second dataset from 2001 to 2020 was used for modeling. Figure 10b shows the normal distribution and Kurtosis and Skewness values. The Kurtosis was closer to 0, which was decreased from 9.77 in the first dataset to 1.35 in the second dataset. The Skewness value displays that the second dataset is symmetrical. Lastly, the third dataset's Kurtosis and Skewness values were calculated. Figure 10c shows that the dataset has the asymmetry of the distribution and symmetry of the dataset. The result has the Kurtosis of 0.03 and the Skewness of 0.12.

The next step, the probability plot, was conducted in Figure 11. Figure 11a shows that the first dataset was far from the normal probability plot and had a high standard deviation (std) of 26.69, indicating that the dataset had significant errors. Figure 11b,c shows an excellent inline of the probability plot for the second and third datasets. The standard deviations decreased to 14.36 and 12.63, respectively.

The next step was the RNN model development. Figure 12 shows the train and validation error of the training and testing of the third dataset. The model performed well in the sample fit. The model fit can be ended when the train and test error is low and close to each other. The figure shows two lines of train results. The test errors were very close to each other.

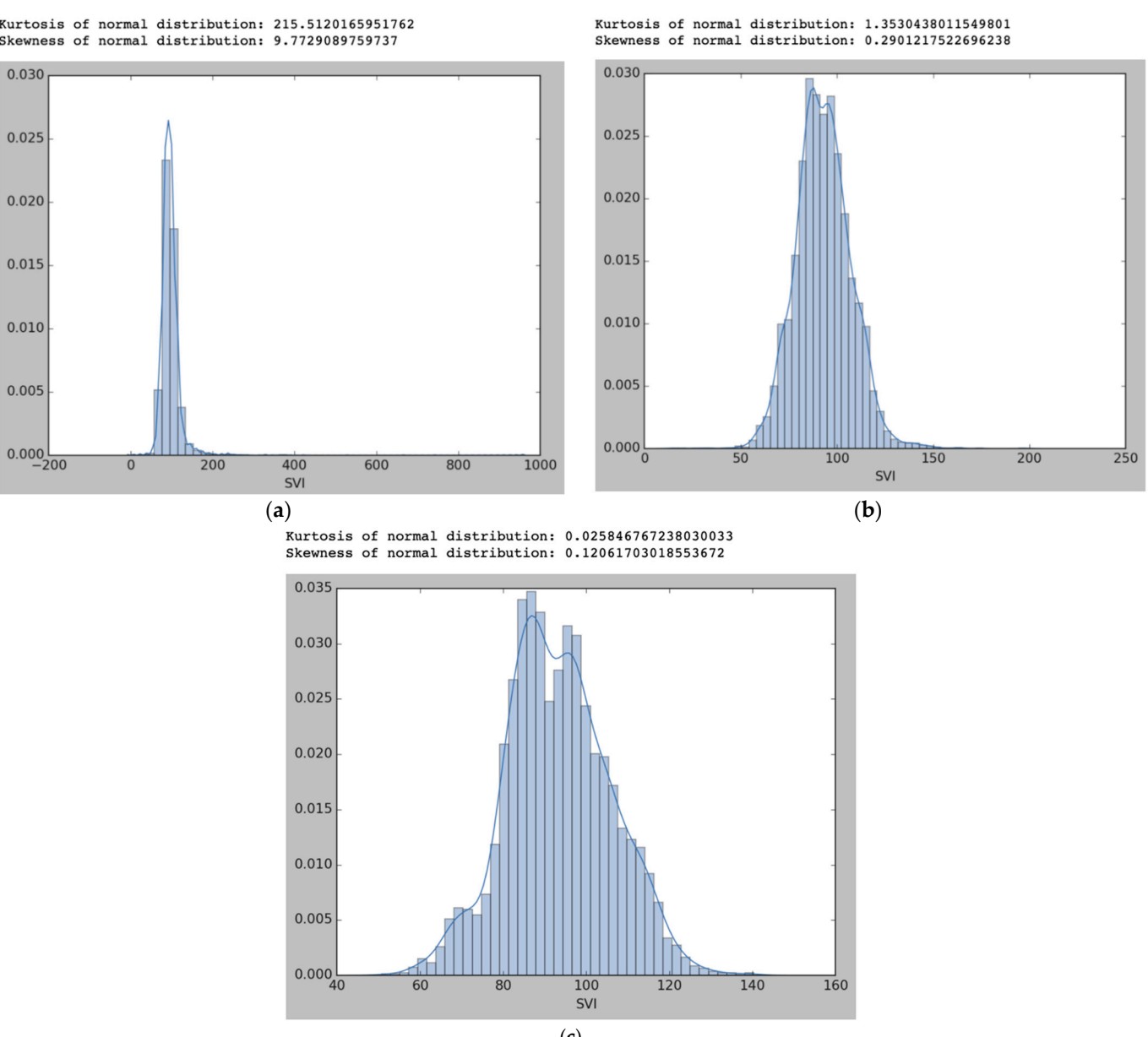

**Figure 10.** This is a figure of normal distribution. (**a**) Normal distribution of the first dataset from 1996 to 2020; (**b**) Normal distribution of the second dataset from 2001 to 2020; (**c**) Normal distribution of the second dataset from 2010 to 2020.

Figures 13–15 show the prediction results of the SVI model. The blue line is the original SVI data, and the green line is predicted. Root Mean Square Error (RMSE) and Mean Absolute Error (MAE) are the most common metrics used to measure the accuracy of a model. RMSE is a quadratic scoring rule that measures the magnitude of errors, i.e., the square root of the average of squared differences between the prediction and original values. MAE is used to measure the average magnitude of the errors in the predictions. It calculates the absolute differences between the prediction and actual values over the test data where all individuals have equal weight. Therefore, both MAE and RMSE can be used to express average model prediction. The metrics range from 0 to $\infty$.

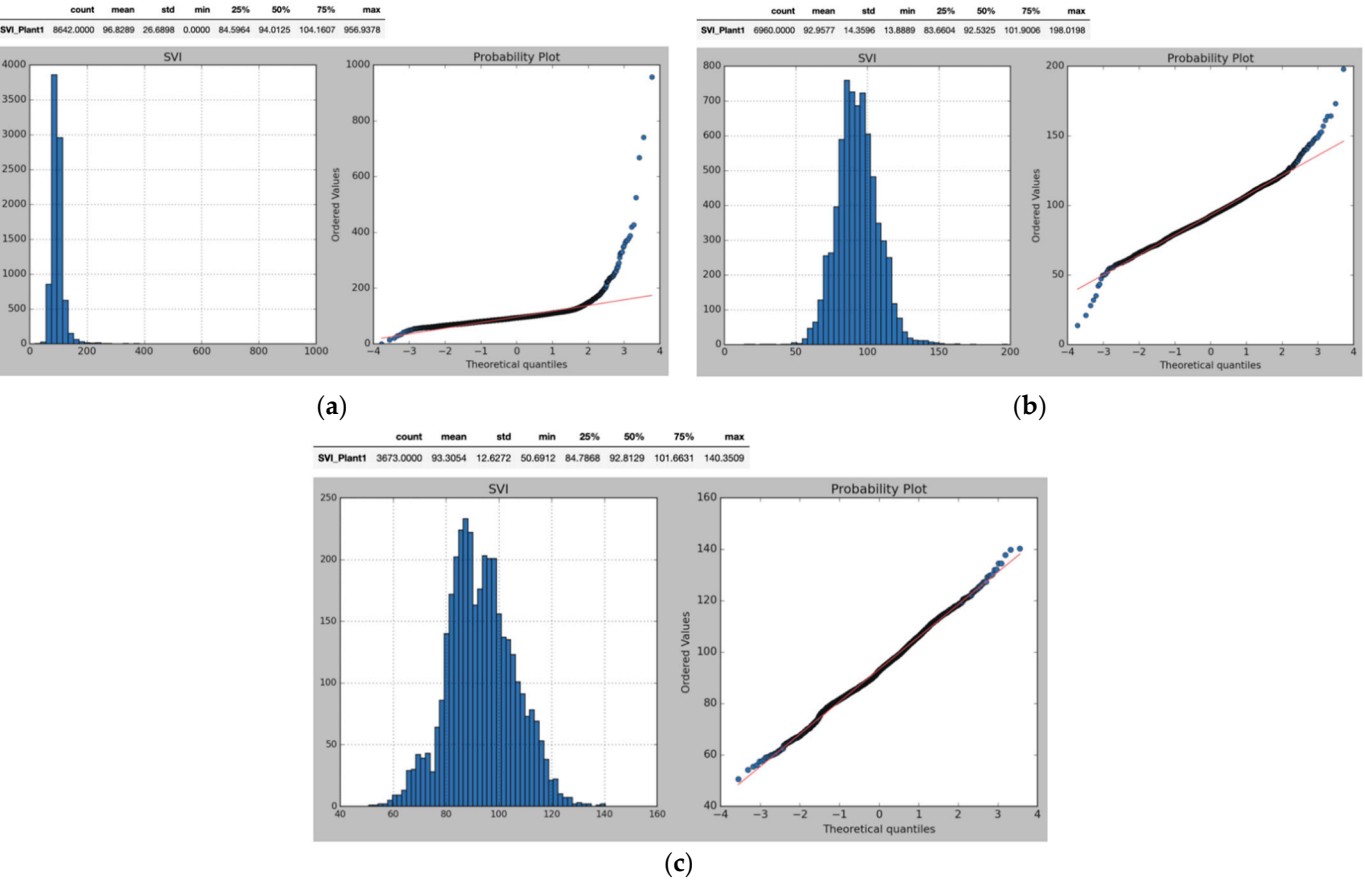

**(a)**

**(b)**

**(c)**

**Figure 11.** This is a figure of a normal probability plot. (**a**) Normal probability plot of the first dataset from 1996 to 2020; (**b**) Normal probability plot of the second dataset from 2001 to 2020; (**c**) Normal probability plot of the third dataset from 2010 to 2020.

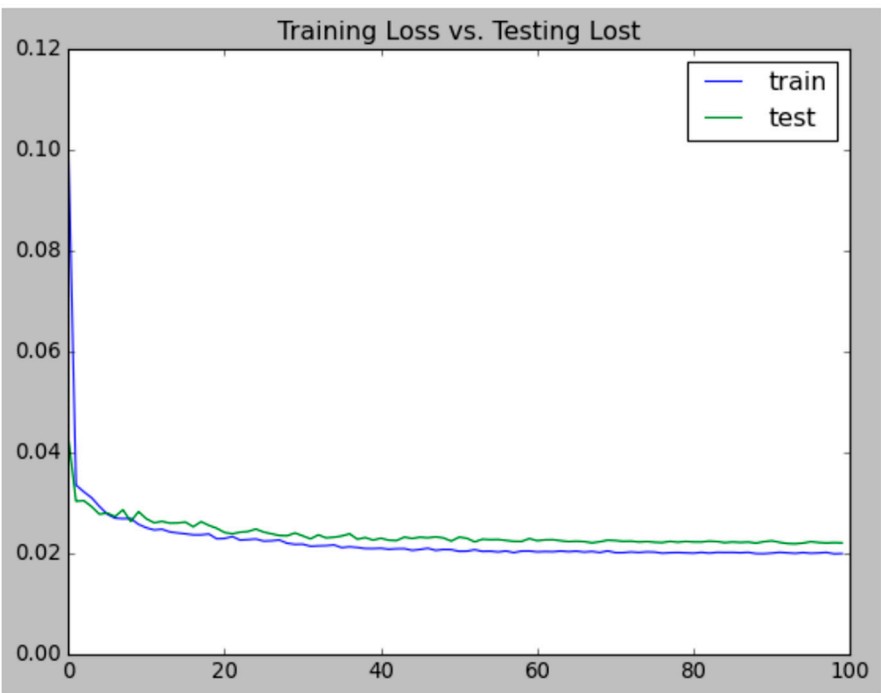

**Figure 12.** Mean Absolute Error between each epoch of the RNN model.

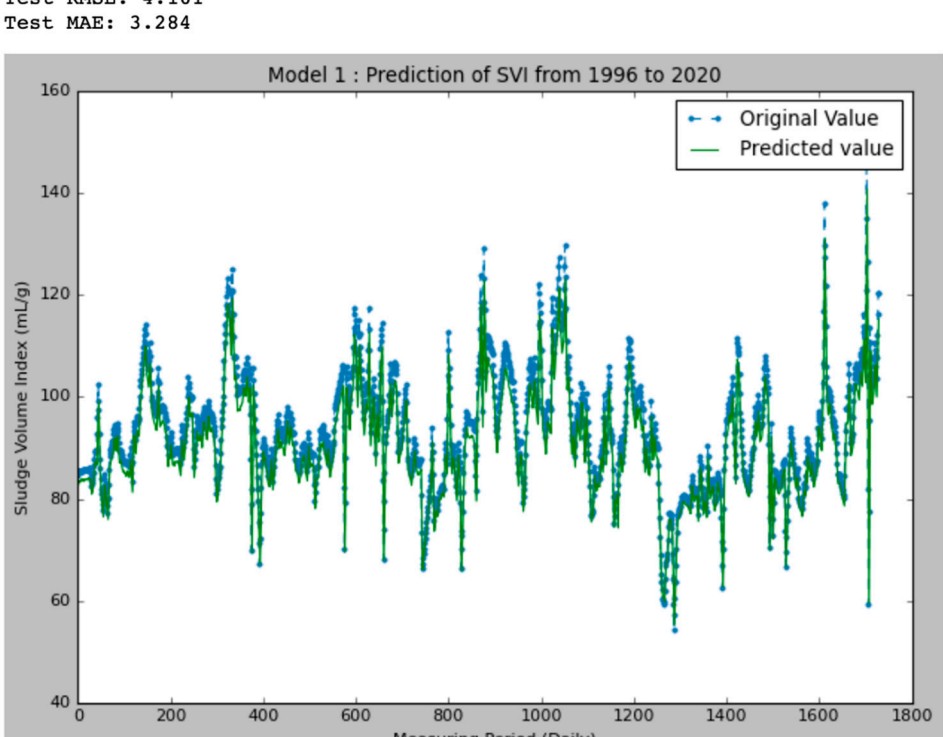

**Figure 13.** The Sludge Volume Index prediction model of the first set of data (1996 to 2020).

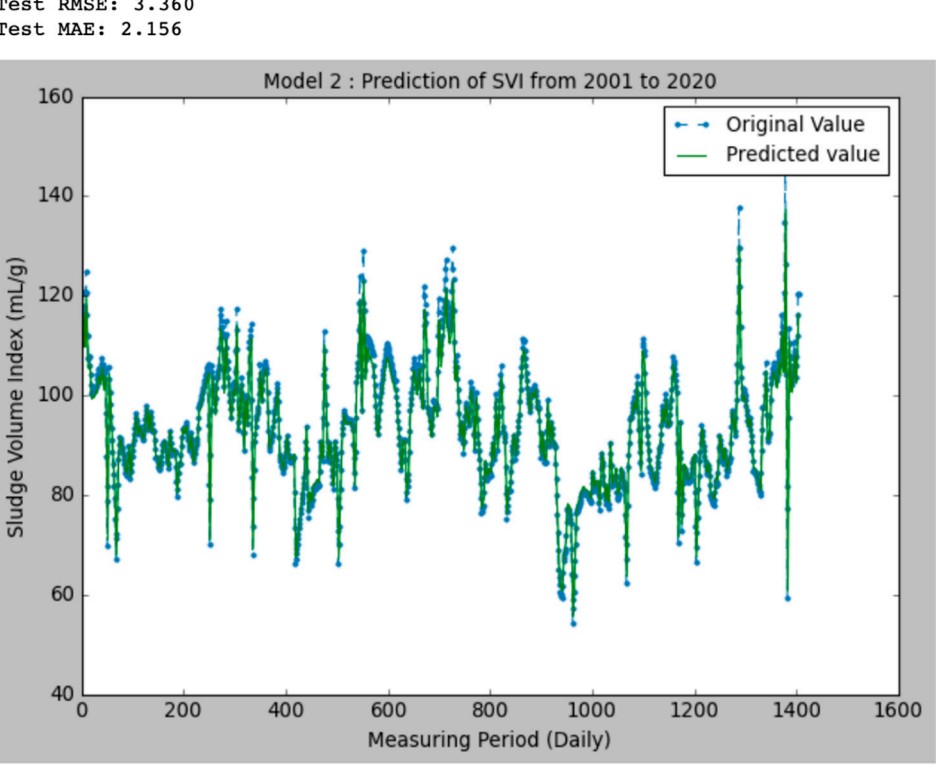

**Figure 14.** The Sludge Volume Index prediction model of the second set of data (2001 to 2020).

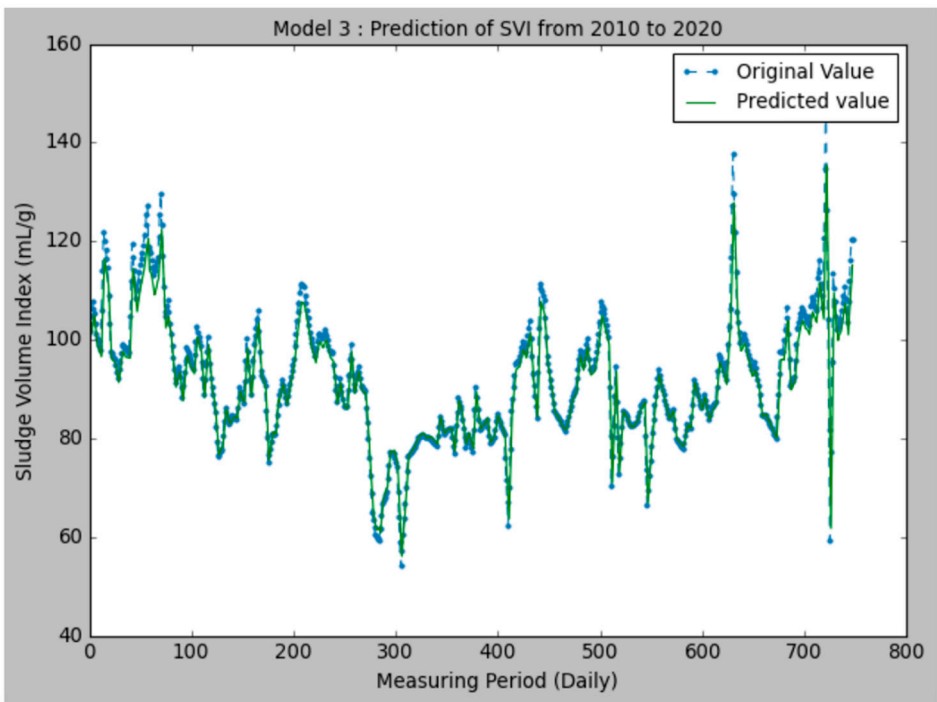

**Figure 15.** The Sludge Volume Index prediction model of the third set of data (2010 to 2020).

Figure 13 shows that the prediction of the first set of data. The prediction has a low RMSE of 4.161 and MAE of 3.284. The prediction of the second dataset after data analysis results in lower RMSE (3.360) and MAE (2.156) values in Figure 14. However, Figure 15 shows that even though the data are more stable, the model's RMSE and MAE were similar or slightly higher than the previous model. Thus, the RNN model can perform well even if the data fluctuate. Data visualization and analysis can help determine the error in the dataset and the system's poor performance.

The next step is to interpret the prediction result of the models. Figures 16–18 show the explainable function applied to the first prediction models. Figure 16 shows that organic loading, $BOD_5$, and flow rate are the most impact input parameters to the SVI prediction, followed by TP, TKN, TSS, and $NH_3N$. Figure 17 shows that when SVI is 114.6, organic loading and flow rate lowered the predicted SVI value, and TP, TSS, $NH_3N$, TKN, and $BOD_5$ increased the SVI value. Lastly, the explainable function can help determine input parameters that affect each output value in Figure 18. Lastly, Figure 19 shows the accuracy of the model using Mean Absolute Error (MAE) for training and testing set. The figure shows the lines are very low and close to each other, which means the model has a good performance. The most important finding is that the organic loading and TP in mass (concentration × flow rate) affect the SVI value most, implying that the WWTP might not be able to supply a proper amount of oxygen in response to the condition change. Thus, the real-time aeration control is thought to achieve a stable SVI.

Similar to the second and third models, organic loading, $BOD_5$, and flow rate were the most related parameters to SVI prediction, followed by TKN, TP, $NH_3N$, and TSS. Depending on the operation condition, the principal parameters affecting the prediction varied. Therefore, applying this explainable function along with model prediction would assist the WWTP operation by closely monitoring the system, visualizing and controlling the system, making a model prediction, interpreting the result, and providing a faulty alarm.

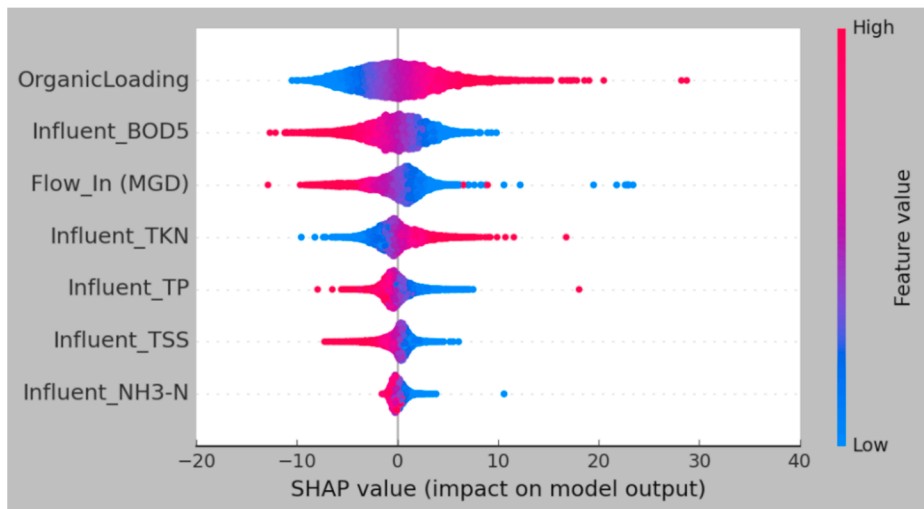

**Figure 16.** The Shapley summary plot of the first model.

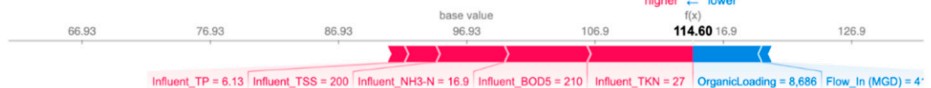

**Figure 17.** The force plot of the first observation of the first model.

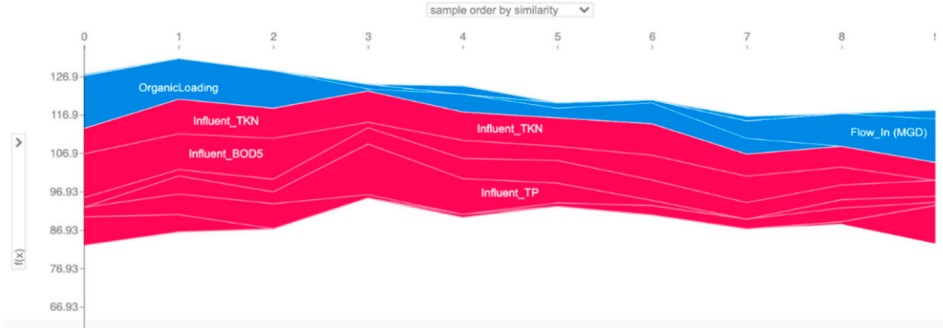

**Figure 18.** The collective force plot of all input variables.

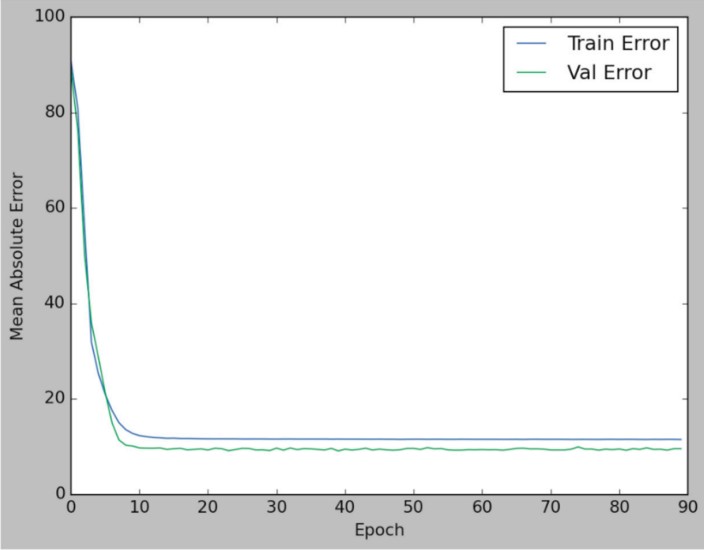

**Figure 19.** Mean Absolute Error between each epoch of the Explainable Function.

## 4. Discussion

The results showed the model's ability to predict SVI despite significant fluctuation in the dataset. The RMSE was ~3 and MAE was ~2 in the SVI range from 50 to 150. However, the RMSE and MAE are still not low enough to conclude that a model is good because it is scale-dependent. A good model can be determined by looking at training and testing errors. The result shows that the second and third models were an excellent in-sample fit, associated with low error measures.

The Explainable Artificial Intelligence (AI) algorithm was useful in explaining the causes of high SVI values. The result can be interpreted in the graph and shows what causes SVI higher or lower for each instance and what the most related parameters to output prediction are. However, the validation of the explainable algorithm needs to be further evaluated with other models. Therefore, it can be concluded that an RNN model with explainable AI can predict SVI and give an operator a suggestion such as which parameters affect higher SVI under widely varying daily conditions.

For the Nine Spring WWTP dataset, organic loading, influent $BOD_5$, and flow rate most affected SVI prediction. It can be caused by oxygen control in an aeration system because aeration impacts $BOD_5$ and SVI. Therefore, aeration control should be thoroughly monitored in an aeration system. Although it was possible to determine which parameter(s) caused higher SVI, reasons and corrective measures must be investigated further for individual WWTPs. The developed method can be applied to other WWTPs, but causative reasons may differ depending on the treatment process, characteristics of raw wastewater, air supply system, DO setpoint and control method, etc., suggesting different solutions for higher SVI.

## 5. Conclusions

Sludge Volume Index or SVI is the most important operational parameter to determine the solid separation potential in the activated sludge system. The RNN model for SVI prediction with data analysis and an Explainable AI function was found to be suitable for WWTP operation. The first step is to collect the data from the WWTP, followed by the data analysis and visualization to see the data pattern. In this study, the data were collected from 1996 to 2020. After data analysis, the appropriate datasets can be created. The data were separated into three sets for training the RNN models. The first set of data from 1996 to 2020 has a standard deviation of 26.69, which is caused by the errors in the dataset. After training the model by using this dataset, the prediction model has an Root Mean Square Error (RMSE) of 4.161 and Mean Absolute Error (MAE) of 3.284. The second dataset is from 2001 to 2020, which has some variations of the activated sludge system. The models can perform better with the RMSE of 3.360 and MAE of 2.156, which is similar to the third dataset, the data from 2010 to 2020. The training and testing error (Figure 12) shows that the RNN model has very low error and the lines are close to each other. Similar to Figure 19 using Shapley explainable function shows that the model has a good performance because of low errors and train and test error lines are near one another. It can be concluded that the RNN architecture can handle normal variations in the activated sludge system well and data analysis is one of the most crucial steps to select the relevant data. Lastly, Shapley interpretation was applied to explain the prediction result. From the dataset, SVI was found to be affected most by organic loading and, thus, influent $BOD_5$ and flow rate. Therefore, it is recommended to improve the aeration control system.

**Author Contributions:** Conceptualization, J.K.P.; formal analysis, P.W.; investigation, P.W.; methodology, P.W.; software, P.W.; supervision, J.K.P.; validation, P.W.; visualization, P.W.; writing—original draft, P.W.; writing—review and editing, J.K.P. All authors have read and agreed to the published version of the manuscript.

**Funding:** This research received no external funding.

**Institutional Review Board Statement:** Not applicable.

**Informed Consent Statement:** Not applicable.

**Conflicts of Interest:** The authors declare no conflict of interest.

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
