# Peer review of "Prediction of Sludge Volume Index in a Wastewater Treatment Plant Using Recurrent Neural Network"

_sustainability, doi:10.3390/su14106276_

Round 1

Reviewer 1 Report

The manuscript can be published

Author Response

The manuscript can be published

Thank you very much.

Reviewer 2 Report

Summary and general comments

In this study, the authors run a supervised machine learning algorithm (recurrent neural network) on activated sludge dataset to predict the sludge volume index.

There are many minor errors in this manuscript, and I suggest authors to correct them. There are many sections that remain unclear. Please see my comments and use whichever you see fit to improve your manuscript.

Specific comments

  1. The title should be specific. There are currently many publications involving machine learning algorithms and to use a general term such as “Artificial intelligence” is highly discouraged. I suggest the authors to include “recurrent neural network” in the title.
  2. The rule of mentioning biological species is not followed. “ natans and M. parvicella” must be mentioned in full, and as an abbreviated form thereafter.
  3. Rules of abbreviation not followed. These errors were detected throughout the manuscript. There are words/terms with the abbreviation re-defined every few paragraphs. Do look for an online resource to properly use abbreviations in scientific writing.
  4. Line 123. There is no context given to these models “ASM1, ASM2, ASM2d, and ASM3”. It is not meaningful to list the model without defining what they are.
  5. Line 172. Some of these parameters may not be obvious to readers and should be defined at first use.
  6. The y-axis of Figure 4 can be limited to 500. The data for the Year 2000 contains many outliers and can be mentioned in the figure’s caption
  7. Are there any meaningful insights for Fig5 and Fig 7? The figure captions and text did not give much information apart from knowing the min-max range of SVI. Consider moving these figures to the supplementary section. In my opinion, a text description is sufficient if the min-max values are the only information obtained from these graphs.
  8. Line 198-200. To avoid overfitting, perhaps a cross-validation method is much preferred than an 80:20 split. Was the 80:20 split done manually? How is the performance of the RNN evaluated? Is the RNN model optimised? These should be mentioned clearly in the methodology section
  9. How was the data transformed to values between 0 and 1. What equation is used? There are many data transformation methods, and this should be conveyed clearly.
  10. What software was used for running the RNN?
  11. It is not clear in the text what helpful insight can be obtained Figs 14-16. What are “high” and “low”? Prediction value above and below the data?

Author Response

Thank you very much. 

Round 2

Reviewer 2 Report

Article is acceptable after this round of revision

This manuscript is a resubmission of an earlier submission. The following is a list of the peer review reports and author responses from that submission.

Round 1

Reviewer 1 Report

The manuscript “Prediction of Sludge Volume Index (SVI) in a Wastewater Treatment Plant Using Artificial Intelligence” is interesting and organized but some minor changes should be addressed:

  1. Please remove the reference from the abstract.
  2. Please update the state-of-the-art by inserting newest references. It will be interesting to discuss about the potential application of sewage sludge (please see https://doi.org/10.3390/app11157139) and about the thermochemical decomposition of sewage sludge (please see https://doi.org/10.37358/RC.20.10.8361).
  3. I recommend to use add a table for the data from “table 1” and “table 2”.
  4. Please use white background for figures and remove the gridlines.
  5. Please extend the conclusion section.
  6. Please check the references and rewrite them according to journal instructions.

Author Response

Thank you very much for your comments and suggestions.

I have attached the revised file here and the point-by-point responses to your comments are as follow: 

1. Please remove the reference from the abstract.

- I removed the reference from the abstract and put it in the introduction

2. Please update the state-of-the-art by inserting newest references. It will be interesting to discuss about the potential application of sewage sludge (please see https://doi.org/10.3390/app11157139) and about the thermochemical decomposition of sewage sludge (please see https://doi.org/10.37358/RC.20.10.8361).

- Update new references

3. I recommend to use add a table for the data from “table 1” and “table 2”.

- Table 1 and table 2 are created from the Python program

4. Please use white background for figures and remove the gridlines.

- Changed to white background and remove the gridlines for figures

5. Please extend the conclusion section.

- Extended the conclusion

6. Please check the references and rewrite them according to journal instructions.

- Checked and corrected the references

Reviewer 2 Report

The authors must improve the introduction and conclusion alot

Author Response

Thank you very much for your suggestions. I really appreciate it.

I improved the introduction and conclusion as your comment.

I have attached the revised file here.  

Reviewer 3 Report

I didn’t see too much scientific contribution in this work. The authors use RNN to predict the time-series SVI. Using Shapley value to explain the result is meaningful, but the idea is too simple and not new, and the results are not verified with other methods, which makes the work unconvincing.

  • The method section: the authors provide the whole process to determine the datasets, which is subjective, without any theory support. In addition, Figure 1 to Figure 6 is redundant, providing the repeat information. In addition, apart from the dataset selection part, the author fails to introduce the details of the core method, i.e. RNN and Shapley function, which is necessary.
  • The discussion section: The author doesn’t provide any further discussion beyond the research results.
  • The conclusion section: Only experiment steps were described in this section.

Other minor problems include the ignore of the explanation of some abbreviations (e.g. WWTP), and all Tables and Figures are not consistent with the scientific format.

Author Response

Thank you very much for your suggestions. I have attached the revised file here and the point-by-point response to your comments are as follow: 

1. The method section: the authors provide the whole process to determine the datasets, which is subjective, without any theory support. In addition, Figure 1 to Figure 6 is redundant, providing the repeat information. In addition, apart from the dataset selection part, the author fails to introduce the details of the core method, i.e. RNN and Shapley function, which is necessary.

Removed a figure and addressed RNN and Shapley function in the introduction section

2. The discussion section: The author doesn’t provide any further discussion beyond the research results.

Added further discussion

3. The conclusion section: Only experiment steps were described in this section.

Improved conclusion section

4. Other minor problems include the ignore of the explanation of some abbreviations (e.g. WWTP), and all Tables and Figures are not consistent with the scientific format.

Add the explanation of WWTPs – Wastewater Treatment Plants

RNN – Recurrent Neural Network

Improved figures to be the same format, but tables were from Python program function. It cannot be changed.

Round 2

Reviewer 2 Report

The authors

Must improve the introduction and also conclusions

The introduction must shows the importance of the work and also more citations from previous studies is needed. Conclusion must be scientifically shows the results is proven and deep interpretations are needed. The authors must use the new references.

Reviewer 3 Report

Thank the authors for the effort to improve the paper. However, the authors didn't respond to my main concern, which is :

The authors use RNN to predict the time-series SVI. Using Shapley value to explain the result is meaningful, but the idea is too simple and not new, and the results are not verified with other methods, which makes the work unconvincing.

I cannot be convinced by work without any validation.